# Completion of FLOT Therapy, Regardless of Tumor Regression, Significantly Improves Overall Survival in Patients with Esophageal Adenocarcinoma

**DOI:** 10.3390/cancers14041084

**Published:** 2022-02-21

**Authors:** Björn-Ole Stüben, Jakob Stuhlfelder, Marius Kemper, Michael Tachezy, Tarik Ghadban, Jakob Robert Izbicki, Carsten Bokemeyer, Marianne Sinn, Karl-Frederick Karstens, Matthias Reeh

**Affiliations:** 1Department of General, Visceral and Thoracic Surgery, University Medical Center Hamburg-Eppendorf, Martinistraße 52, 20246 Hamburg, Germany; b.stueben@uke.de (B.-O.S.); jakob.stuhlfelder@stud.uke.uni-hamburg.de (J.S.); m.kemper@uke.de (M.K.); m.tachezy@uke.de (M.T.); t.ghadban@uke.de (T.G.); izbicki@uke.de (J.R.I.); k.karstens@uke.de (K.-F.K.); 2Department of Internal Medicine II/Oncology/Hematology/BMT/Pneumology, Center of Oncology, University Hospital Hamburg-Eppendorf, 20246 Hamburg, Germany; c.bokemeyer@uke.de (C.B.); ma.sinn@uke.de (M.S.)

**Keywords:** esophageal adenocarcinoma, FLOT therapy, tumor regression grading

## Abstract

**Simple Summary:**

Multimodal therapy concepts, including surgery and systemic therapy, are the mainstay in the treatment of esophageal adenocarcinoma. Despite the technical advances in the surgical field and the widespread use of chemoradiation therapy, the prognosis and overall survival for esophageal cancer remains poor. Therapy with a continuous infusion of 5-FU, leucovorin, oxaliplatin, and docetaxel (FLOT protocol) has been shown to improve the overall survival for patients with esophageal adenocarcinoma. However, uncertainty exists as to whether patients with poor tumor responses should complete the chemotherapy following surgery. The aim of our study was to analyze the effect of neoadjuvant and adjuvant FLOT therapy on the outcomes of patients following an esophagectomy for adenocarcinoma, with the focus on the tumor regression grading and the completion of the systemic therapy. We showed that the completion of the systemic therapy, regardless of the tumor regression grading, improved the outcomes of patients with esophageal adenocarcinoma. Subgroup analyses further showed that complications from chemotherapy reduced the overall survival, while surgical complications did not.

**Abstract:**

Esophageal cancer is the eighth most common cancer worldwide, with poor prognosis and high mortality. The combination of surgery and systemic therapy provide the best chances for long-term survival. The purpose of this study was to analyze the impact of the FLOT protocol on the overall survival of patients following surgery for esophageal adenocarcinoma, with a focus on the patients who did not benefit in terms of pathological remission from the neoadjuvant therapy. A retrospective analysis of all the patients who underwent esophagectomies from 2012 to 2017 for locally advanced adenocarcinomas of the esophagus at a tertiary medical center was performed. The results show that the completion of systemic therapy, regardless of the tumor regression grading, had a significant positive impact on the overall survival. The patients with complete regression and complete systemic therapy showed the best outcomes. Anastomotic insufficiency did not negatively impact the long-term survival, while complications of the systemic therapy led to significantly reduced overall survival. We conclude that adjuvant systemic therapy should, when possible, always be completed, regardless of the tumor regression, following an esophagectomy.

## 1. Introduction

Esophageal cancer is the eighth most common cancer worldwide [1]. The condition has a poor prognosis, with five-year overall survival rates of 15–20% [2]. Squamous cell and adenocarcinoma are the main subtypes of esophageal malignancies, and they make up 95% of the cases. Adenocarcinomas have been increasing in recent years, and they are the most common subtype in industrialized nations [3]. Obesity, gastroesophageal reflux, and smoking are the main known risk factors for the development of this tumor entity [4,5,6]. The preoperative staging is vital for selecting the appropriate therapeutic approach for patients with esophageal adenocarcinomas. In patients with advanced regional-stage malignancies (Stage III and above), both chemotherapy and chemoradiotherapy, followed by esophagectomy, is recommended [7,8,9,10]. However, controversy exists as to which perioperative therapy is superior [11,12,13]. Whilst radiotherapy relies mainly on local/locoregional disease control, systemic chemotherapy potentially eliminates micrometastases and, in theory, reduces the risk of metastasis following surgery. Several landmark studies show the overall survival benefits for patients with esophageal malignancies when receiving neoadjuvant or perioperative chemoradiation or chemotherapy [14,15,16,17]. The CROSS trial, which analyzed 366 patients, showed an improvement in the overall survival after neoadjuvant chemoradiation with 41.4 Grey (Gy) and concomitant weekly carboplatin and paclitaxel, with subsequent surgical resection, compared to surgery alone (esophageal adenocarcinoma: a three-year survival of 55% vs. 46%, *p* = 0.049) [14]. Carrying on from this, different protocols of perioperative chemotherapy have been applied as an alternative to neoadjuvant chemoradiation [15,16,17]. Two multicenter trials, the MAGIC trial [15] and the ACCORD trial [16], found significant increases in the overall survival for perioperative chemotherapy plus surgery compared to surgery alone for cisplatin-fluoropyrimidine-based therapy regimes. The results of the MAGIC trial have recently been challenged by the presentation of the results from the AIO-FLOT-4 trial [17]. This randomized phase III study compared the perioperative chemotherapy regimens, FLOT (5-FU, leucovorin, oxaliplatin, and docetaxel) and ECF (epirubicine, cisplatin, and 5-FU). The overall survival significantly improved (median survival: 35 vs. 50 months; projected five-year-survival: 36 versus 45 months) for patients treated with FLOT. However, more than 40% of the patients treated with perioperative FLOT showed either minimal or no pathological response to this treatment, according to Becker et al. [18]. In addition, almost 30% of the patients treated with perioperative chemotherapy experienced serious adverse events [17]. Thus, only 65.6% of the patients analyzed by Cunningham et al. began postoperative chemotherapy and, of these patients, only 75.9% completed the full regime of postoperative chemotherapy [15]. Hence, it remains unclear whether patients with minimal or no tumor regression (as shown in the postoperative histopathological findings) benefit from the completion of the chemotherapy regimen. The aim of this study was to investigate the impact of the completion of perioperative chemotherapy on the overall survival in patients with esophageal adenocarcinoma with regard to the tumor regression grading, according to Becker et al. [18]. Furthermore, the impact of the adverse events due to chemotherapy and surgical complications on the overall survival were investigated.

## 2. Methods

### 2.1. Patient Selection, Study Design, and Inclusion Criteria

A total of 483 patients with esophageal cancer were treated surgically from 2012 to 2017 at our department. Only patients with histologically confirmed advanced adenocarcinomas who subsequently received oncologic esophagectomies were selected. Patients with distant metastases and patients treated with radiotherapy were excluded. In the surgery-only group, patients with early carcinomas (pT1, pT2, and pN0) were excluded for better comparability, since only patients with locally advanced carcinomas were considered for perioperative chemotherapy. Hence, 124 patients were included in the study. A total of 53 (42.7%) patients received surgery alone, while 71 (57.3%) patients received perioperative chemotherapy, according to the FLOT protocol. The follow ups were performed by consulting the electronic medical records of our institution if the patients were last treated here. If this was not the case, the local health care authority records were consulted, and, in case this was inconclusive, the General Practitioner of the patient was contacted by phone.

### 2.2. Statistical Analysis

The patients treated by surgery alone and the patients treated with surgery and chemotherapy (FLOT) were compared in terms of the overall survival. Different subgroup analyses were then performed. The group with chemotherapy was divided into two groups: those who completed the neoadjuvant and adjuvant chemotherapy (receiving all eight planned cycles) vs. those who completed neoadjuvant chemotherapy only. The chemotherapy collective was further divided into groups with “good pathological response” and groups with “minimal or no pathological response”, according to the histopathological tumor regression grading system of Becker et al. [18]. The effects of the adverse events of the chemotherapy and surgical complications were also analyzed. Therefore, the medical records were screened for the common side effects of the chemotherapy and complications on an in-patient basis, which were classified according to Dindo et al. [19]. The primary endpoint for all of the subgroup analyses was the overall survival. All of the patients were included in the survival analyses.

The average values are given as medians, with minimums and maximums as range. If the median values were not applicable, then the mean values were used. The survival intervals were computed from the time of the surgery to the time of disease-related death. A group comparison was performed using a chi-square test. Kaplan–Meier curves were constructed to estimate the survival, and a log-rank test was performed to compare the survival variables in the univariate analysis. The data was analyzed using the SPSS software (Version 25). All of the tests used were two-sided. The cut-off for the significance level was set at 0.05.

### 2.3. Statement of Ethics

The study was approved by the Medical Ethical Committee (approval number: PV3548), Hamburg, Germany. Informed consent was obtained from all patients before study inclusion. All of the procedures performed in this study involving human participants were in accordance with the ethical standards of the Institutional and National Research Committee, and with the 1964 Helsinki declaration and its later amendments, or comparable ethical standards.

## 3. Results

### 3.1. Clinical Data

In total, 124 patients (22 female and 102 male), with a median age of 63 years (range 27–88 years), were included in the analysis. All of the patients received either open (66.1%; *n* = 82), hybrid (7.2%; *n* = 9), or total (26.6%; *n* = 33) laparoscopic Ivor-Lewis esophagectomies, with radical two-field lymphadenectomies. A total of 53 (42.7%) patients received surgery alone, while 71 (57.3%) patients received surgery and chemotherapy (FLOT). Of the latter group, 57 of 71 (80.3%) patients received the full (neoadjuvant and adjuvant) FLOT therapy regimen perioperatively, while 14 (19.7%) patients did not complete the adjuvant chemotherapy following surgery. In the subgroup analysis, the 14 patients with incomplete adjuvant chemotherapy showed no significant differences with regard to age, the ASA score, the Clavien–Dindo score, or the anastomotic leakage rate (data not shown).

When comparing the clinical data, a significant difference between the surgery-only group and the surgery-plus-chemotherapy (FLOT) group with regard to the T and N categories was found (*p* = 0.020 and *p* = 0.002, respectively), which indicates that there were less locally advanced tumors after chemotherapy in the latter group. In a subgroup comparison between the patients with good or minimal/no pathological response to the FLOT therapy and the surgery-only group, these differences remained significant in patients with good responses (*p* < 0.0001 and *p* < 0.0001, respectively). The patients with minimal or no pathological response not only did not show less locally advanced tumors, but also demonstrated reduced lymph node involvement (*p* = 0.011). Severe complications, according to the Clavien–Dindo classification, were observed more frequently in the surgery-only group, compared to the perioperative FLOT group (*p* < 0.0001). This difference remained significant in the subgroup analyses that compared the patients with good and minimal or no response against the surgery-only group (*p* = 0.009 and *p* < 0.0001). Of note, significantly more patients with good response to the FLOT therapy received total laparoscopic esophagectomies, as compared to the surgery-only group (*p* = 0.010). Further clinical data are summarized in Table 1.

### 3.2. Surgery Alone vs. FLOT

When comparing all the patients that had received surgery alone to the patients who had received FLOT, without differentiating for therapy discontinuation or the response to therapy, the FLOT group showed longer overall survivals, with a median of 15.5 months (range: 14.6–16.3 months), as compared to the surgery-only group, with a median survival of 14.4 months (range: 12.4–16.4 months). There was no statistically significant difference between the two groups (*p* = 0.307), as is shown in Figure 1a.

We then performed a subgroup analysis of the dataset and analyzed the effect of the completion of the therapy on the overall survival. The patients that had completed the FLOT regimen showed significantly (*p* < 0.001) longer overall survivals, with a median of 15.8 months (range: 9.8–21.8 months), compared to the patients who did not complete the chemotherapy, with a median survival of 8.3 months (range: 6.3–10.4 months). Moreover, the FLOT-completed group showed significantly (*p* = 0.036) improved survivals compared to the patients who had received only surgery, with a median of 14.4 months (range: 12.4–16.4 months). Interestingly, the patients who did not complete the chemotherapy also had significantly (*p* < 0.001) inferior median overall survivals when compared to the surgery-only group, as is depicted in Figure 1b.

### 3.3. Good Response vs. Minimal or No Response

Of the patients treated with FLOT, a “good response” was detected in 19 (26.8%) patients, and “minimal or no response” was detected in 52 (73.2%) patients. The patients in the “good response” group showed a significant (*p* = 0.004) mean overall survival advantage of 52.9 months (range: 34.6–71.3 months) over the “minimal/no response” group, with a mean survival of 25.6 months (range: 16.1–35.2 months), and a significant (*p* = 0.003) improved mean survival, to the surgery-only group, with a median survival of 24.9 months (range: 16.5–33.3 months). No significant survival difference was observed between the “minimal/no response” group and the surgery-only group (*p* = 0.840). The results are shown in Figure 2a.

Of note, the patients who showed “minimal or no response” to the neoadjuvant therapy demonstrated significant (*p* = 0.032) survival benefits, with a mean of 38.1 months (range: 26.9–49.3 months) when completing their FLOT regime following surgery, as compared to the patients who did not finish their chemotherapy (mean survival: 7.6 months; range: 5.4–9.8 months). See Figure 2b. A summary of the reasons for therapy discontinuation are shown in Table 2.

### 3.4. Surgical Complications vs. Chemotherapy Complications

When comparing the surgical complications and adverse events due to the FLOT therapy (see Table 2), we observed that the patients who had complications related to chemotherapy (*n* = 20; 28.2%) had significantly (*p* = 0.040) shorter overall survivals (median: 11.2 months; range: 9.3–13.2 months), compared to the patients who had chemotherapy without adverse events (median: 20.0 months; range: 13.1–26.9 months), as is shown in Figure 3a.

As it is the most severe surgical complication, the anastomotic leakage was analyzed in the subgroup analysis. We showed that this complication did not significantly impact the overall survival (*p* = 0.632). The patients with anastomotic leakage had a median survival of 14.4 months (range: 11.8–17.1 months), compared to the patients without anastomotic leakage, with a median survival of 14.9 months (range: 14.1–15.8 months). See Figure 3b.

## 4. Discussion

The MAGIC trial demonstrated that perioperative chemotherapy with the ECF regimen (epirubicin, cisplatin, and infused fluorouracil) decreased the tumor size and stage, and significantly improved the progression-free and overall survival compared to surgery alone [15]. The results from the AIO-FLOT trial show that the overall survival significantly improved for the patients treated with FLOT compared to those treated with ECF. However, more than 40% of the patients treated with perioperative FLOT showed either minimal or no pathological response to this treatment [17]. In our group, however, 73% of the patients treated with FLOT demonstrated either minimal or no histopathological response. In line with a recently published meta-analysis, our results demonstrate no overall survival benefit for patients receiving perioperative chemotherapy compared to surgery alone, irrespective of them completing the chemotherapy [20]. However, we identified significantly reduced local tumor extension (ypT category) and lymph node involvement (ypN category) in the group treated with FLOT. This might be explained by the achieved downstaging of the local tumor burden by the administered neoadjuvant chemotherapy.

In our cohort, a high number of locally advanced esophageal cancers were found to have received only surgery when, in fact, a perioperative chemotherapy should have been administered. This might be explained by staging errors, despite proper preoperative staging using a CT scan, endoscopy, and an endoscopic ultrasound. Dolan et al. investigated a chemotherapy-naïve cohort of cT2cN0 esophageal cancers and identified understaging in 56% of the patients. Moreover, a spread of the cancer to the regional lymph nodes was detected in the pathological specimens in 52% of the cases, which underlies the difficulty of correct preoperative staging [21]. However, controversy exists as to whether cT2 cN0 esophageal cancers should also be treated by chemotherapy, since many of the cancers are not staged properly [22,23]. From 2021 to 2017, most patients with cT2 and cN0 staged esophageal cancers were treated by surgery only in our clinic.

This retrospective analysis shows no statistically significant difference in the OS for patients with surgery alone versus those receiving perioperative FLOT chemotherapy; however, for the patients achieving complete pathological remission and those completing the adjuvant part of FLOT therapy, there is a clear improvement in the prognoses. Given the abovementioned uncertainty of clinical staging in the determination of T2 vs. T3 and NO vs. N1 stages, our results argue in favor of the broad use of perioperative chemotherapy in most patients, except for T1. Of note, the patients with good response to chemotherapy and who completed the perioperative course of therapy showed the best long-term survival of all the patients. In contrast, the patients in whom the chemotherapy was discontinued had inferior outcomes, even when compared to the surgery-only group. Similar results have been reported in a study comprising 302 patients who received neoadjuvant chemotherapy and radiation for esophageal adenocarcinomas. In this analysis, only patients with good histopathological response (<10% vital residual tumor cells) demonstrated an improved five-year survival rate, which was independent from the surgical approach [24,25]. Interestingly, in our cohort, even patients with minimal or no response to chemotherapy (tumor regression grades 2 and 3) showed an improved overall survival if the chemotherapy was completed following surgery. In fact, only patients who completed the full course of FLOT therapy benefited from the therapy at all. Hence, the completion of the chemotherapy regimen should be achieved whenever possible.

However, complications during FLOT therapy or surgery occur frequently and might alter the course of the perioperative chemotherapy. Thus, we analyzed the treatment-related adverse events. We identified significantly more severe complications in the surgery-only group as compared to the group receiving perioperative FLOT therapy, according to the Clavien–Dindo classification. This might be explained by a more extended or technically difficult surgical approach in the surgery-only groups since the reduction in the local tumor sizes achieved by neoadjuvant chemotherapy might lead to more feasible resections. In addition, more open esophageal resections were performed in the surgery-only group, which are known to be accompanied by a greater number of pulmonary and cardiovascular complications, which necessitate further interventions [26]. Moreover, more patients with higher ASA scores were found in the surgery-only group, which increases the probability of perioperative complications. Of note, 18 patients presented with ASA scores of IV. Of these, a total of 6 patients underwent perioperative chemotherapy, and 12 patients underwent surgery only because of staging inaccuracy or comorbidities, which made chemotherapy impossible. This can be explained by the fact that, as a tertiary center for esophageal cancer treatment, a high number of patients with significant comorbidities are presented. These patients are interdisciplinary when reviewed in-depth with regard to the possibility of neoadjuvant treatment or primary resection.

Additionally, our results indicate that the adverse events of the chemotherapy significantly reduced the overall survival. By contrast, the major surgical complication of anastomotic leakage did not have an impact on the overall survival. Our findings could possibly be explained by the fact that, while surgical complications may postpone adjuvant chemotherapy by prolonging the postoperative stay in hospital, adverse events from chemotherapy may lead to an interruption, a dosage reduction, or even to the discontinuation of the therapy altogether. This is in line with a recent study, which demonstrates, in 1539 patients, that anastomotic leakage did influence the short-term outcome, but not the long-term survival [27].

However, our data are of a retrospective nature, and the investigated groups, especially the patients treated with FLOT who did not receive adjuvant chemotherapy, are relatively small. Moreover, there might be heterogeneity in the patients treated with FLOT therapy, since we only investigated the number of administered cycles of chemotherapy and did not analyze the effect of a dosage reduction. Hence, the results need to be interpreted with caution and further prospective multicenter studies are needed to validate our findings. Of note, the regimes for perioperative chemotherapy are constantly re-evaluated, and the use of immunological agents is rapidly implemented in the treatment of esophageal carcinomas [28,29]. Hence, we might see more individualized systemic treatment in the near future, which might lead to fewer and less severe side effects because of the systemic therapy in general.

## 5. Conclusions

The completion of the perioperative FLOT regime significantly increased the overall survival when compared to surgery alone. The patients with good tumor-regression-grading therapy showed the best overall survival rates. It is vital that FLOT therapy is administered completely perioperatively, and that it is administered in a neoadjuvant as well as an adjuvant setting. Regardless of the tumor regression grading, perioperative FLOT therapy should be continued following surgery. Patients who do not complete the therapy regimen do not benefit from the therapeutic successes seen in patients who complete the FLOT therapy. Even patients with minimal tumor regression show improved overall survival when completing FLOT therapy.

Anastomotic leaks do not have a negative impact on the overall survival. However, adverse events from chemotherapy reduce the overall survival rates, which is possibly due to the fact that they lead to the discontinuation of chemotherapy, which means that the target dosage cannot be achieved.

## Figures and Tables

**Figure 1 cancers-14-01084-f001:**
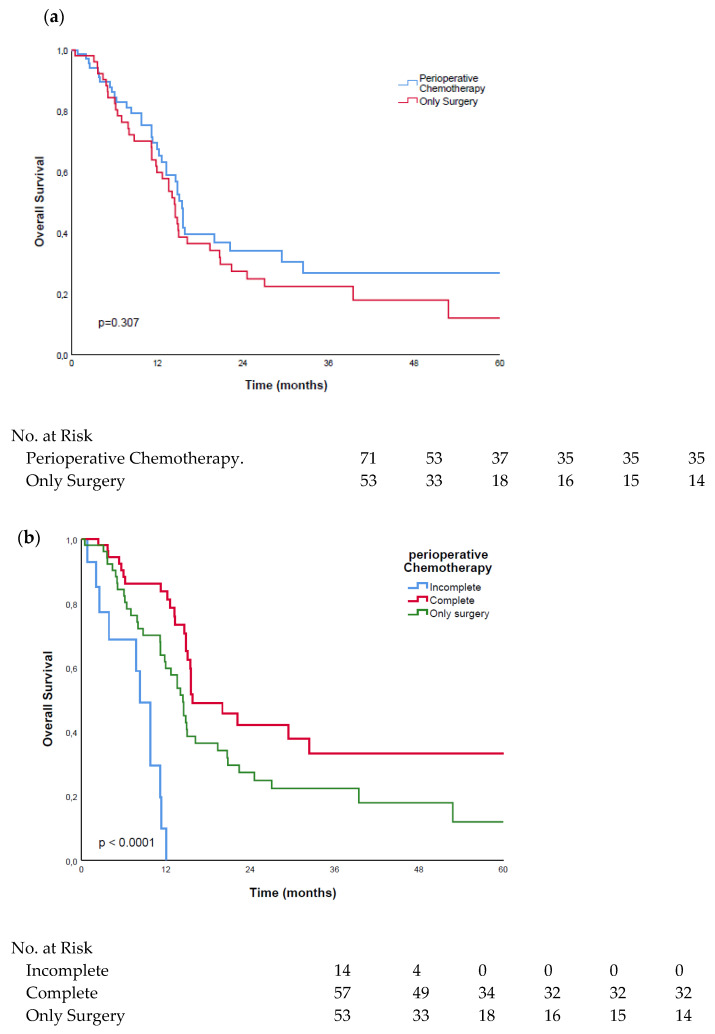
Kaplan–Meier curves of patients treated with FLOT and surgery only: (**a**) survival comparison of patients treated with surgery only, and in combination with perioperative FLOT; (**b**) survival comparison of patients with completed and uncompleted FLOT treatment.

**Figure 2 cancers-14-01084-f002:**
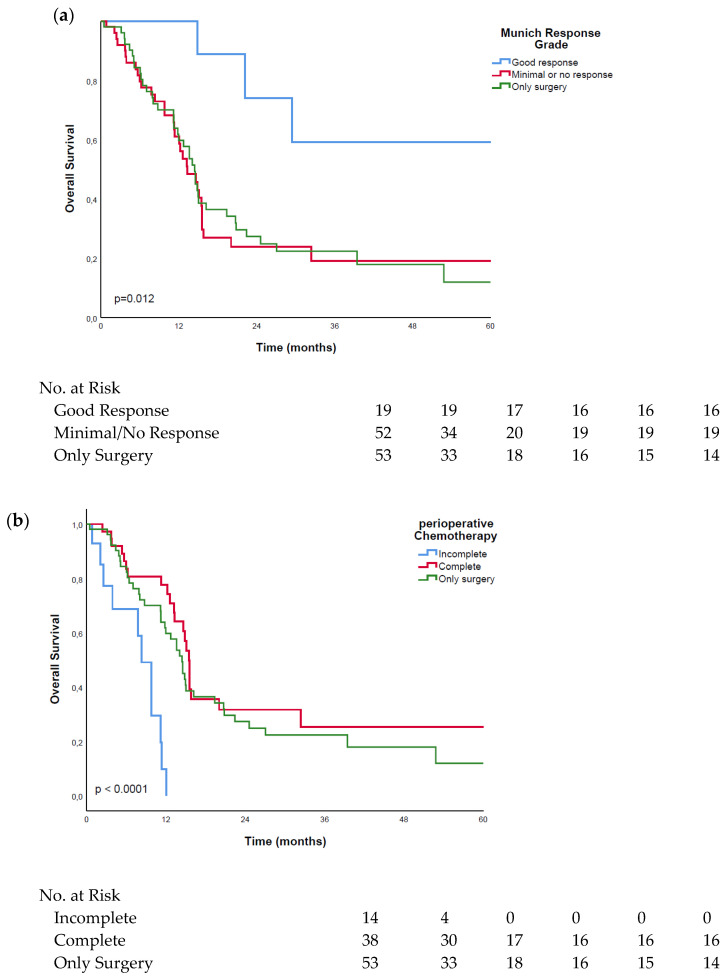
Kaplan–Meier curves of patients with regard to the histopathological response to the FLOT treatment: (**a**) survival comparison of patients with a good and “minimal or no response” to the FLOT treatment; (**b**) survival comparison of patients with a “minimal or no response” to chemotherapy without completion of the full FLOT treatment.

**Figure 3 cancers-14-01084-f003:**
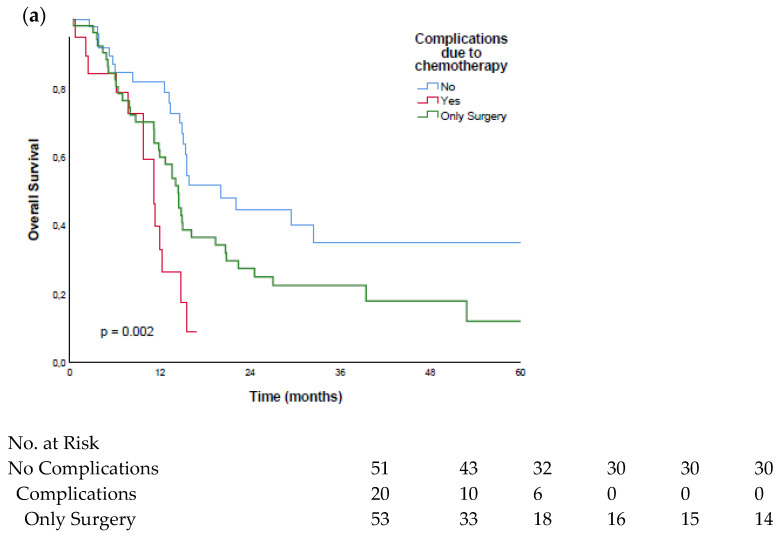
Kaplan–Meier curves of patients with complications related to surgery or chemotherapy: (**a**) survival comparison of patients with complications related to the FLOT treatment; (**b**) survival comparison of patients with anastomotic leakage (AL).

**Table 1 cancers-14-01084-t001:** Summary of clinical data.

Variables.	All	Surgery Only	Perioperative CTx	*p* Value *	Perioperative CTx: Good Response	*p* Value *	Perioperative CTx: Minimal/No Response	*p* Value *
Patients	124 (100%)	53 (42.7%)	71 (57.3%)		19 (26.8%)		52 (73.2%)	
**Age (years)**								
≤60	42 (33.9%)	13 (24.5%)	29 (40.8%)	0.058	7 (36.8%)	0.304	22 (42.3%)	0.053
>60	82 (66.1%)	40 (75.5%)	42 (59.2%)		12 (63.2%)		30 (57.7%)	
**Sex**								
Male	102 (82.3%)	44 (83.0%)	58 (81.7%)	0.848	16 (84.2%)	0.905	42 (80.8%)	0.765
Female	22 (17.7%)	9 (17.0%)	13 (18.3%)		3 (15.8%)		10 (19.2%)	
**T category**								
ypT0	5 (4.0%)	0 (0.0%)	5 (7.0%)	*0.020*	5 (26.3%)	*<0.0001*	0 (0.0%)	*0.136*
yp/pT1	8 (6.5%)	0 (0.0%)	8 (11.3%)		4 (21.1%)		4 (7.7%)	
yp/pT2	21 (16.9%)	9 (17.0%)	12 (16.9%)		6 (31.6%)		6 (11.5%)	
yp/pT3	75 (60.5%)	38 (71.7%)	37 (52.1%)		4 (21.1%)		33 (63.5%)	
yp/pT4	15 (12.1%)	6 (11.3%)	9 (12.7%)		0 (0.0%)		9 (17.3%)	
**N category**								
yp/pN0	29 (23.4%)	5 (9.4%)	24 (33.8%)	*0.002*	12 (63.2%)	*<0.0001*	12 (23.1%)	*0.011*
yp/pN1	37 (29.8%)	22 (41.5%)	15 (21.1%)		6 (31.6%)		9 (17.3%)	
yp/pN2	27 (21.8%)	9 (17.0%)	18 (25.4%)		1 (5.3%)		17 (32.7%)	
yp/pN3	31 (25.0%)	17 (32.1%)	14 (19.7%)		0 (0.0%)		14 (26.9%)	
**R category**								
0	109 (87.9%)	46 (86.8%)	63 (88.7%)	0.743	19 (100.0%)	0.095	44 (84.6%)	0.750
1	15 (12.1%)	7 (13.2%)	8 (11.3%)		0 (0.0%)		8 (15.4%)	
**AEG**								
I	47 (37.9%)	23 (43.4%)	24 (33.8%)	0.285	7 (36.8%)	0.689	17 (32.7%)	0.252
II	71 (57.3%)	29 (54.7%)	42 (59.2%)		11 (57.9%)		31 (59.6%)	
III	6 (4.8%)	1 (1.9%)	5 (7.0%)		1 (5.3%)		4 (7.7%)	
**ASA**								
1	1 (0.8%)	1 (1.9%)	0 (0.0%)	0.081	0 (0.0%)	0.315	0 (0.0%)	*0.201*
2	26 (21.0%)	11 (20.8%)	15 (21.1%)		4 (21.1%)		11 (21.2%)	
3	79 (63.7%)	29 (54.7%)	50 (70.4%)		14 (73.7%)		36 (69.2%)	
4	18 (14.5%)	12 (22.6%)	6 (8.5%)		1 (5.3%)		5 (9.6%)	
**Clavien–Dindo**								
0	25 (20.2%)	2 (3.8%)	23 (32.4%)	*<0.0001*	7 (36.8%)	*0.009*	16 (30.8%)	*<0.0001*
1	5 (4.0%)	1 (1.9%)	4 (5.6%)		0 (0.0%)		4 (7.7%)	
2	32 (25.8%)	10 (18.9%)	22 (31.0%)		4 (21.1%)		18 (34.6%)	
3	24 (19.4%)	16 (30.2%)	8 (11.3%)		3 (15.8%)		5 (9.6%)	
4	36 (29.0%)	22 (41.5%)	14 (19.7%)		5 (13.9%)		9 (17.3%)	
5	2 (1.6%)	2 (3.8%)	0 (0.0%)		0 (0.0%)		0 (0.0%)	
**Type of surgery**								
Open	82 (66.1%)	39 (73.6%)	43 (60.6%)	0.100	8 (42.1%)	*0.010*	35 (67.3%)	0.408
Laparoscopic	33 (26.6%)	9 (17.0%)	24 (33.8%)		10 (52.6%)		14 (26.9%)	
Hybrid	9 (7.3%)	5 (9.4%)	4 (5.6%)		1 (5.3%)		3 (5.8%)	
**Anastomotic leak**								
No	98 (79.0%)	42 (79.2%)	56 (78.9%)	0.960	13 (68.4%)	0.341	43 (82.7%)	0.653
Yes	26 (21.0%)	11 (20.8%)	15 (21.1%)		6 (31.6%)		9 (17.3%)	

Patients with distant metastases (M1) were excluded. Grading was only available in the surgery-only group (G2: *n* = 17 (32.1%); G3: *n* = 36 (67.9%)). Significant values are highlighted in italic. * in comparison to the surgery-only group.

**Table 2 cancers-14-01084-t002:** Causes of FLOT complications and terminations.

Type of Complication	Complications during CTx *	Cause for Discontinuitation of CTx *
	*n* = 20 of 71 (28.2%)	*n* = 14 of 71 (19.7%)
Weight loss	2 (7.7%)	0 (0.0%)
Nausea and vomiting	4 (15.4%)	2 (12.5%)
Polyneuropathy	4 (15.4%)	2 (12.5%)
Pneumonitis	3 (11.5%)	3 (18.8%)
Fatigue	4 (15.4%)	3 (18.8%)
Diarrhea	3 (11.5%)	2 (12.5%)
Edema of lips and neck	1 (3.8%)	1 (6.3%)
Hand foot syndrome	1 (3.8%)	1 (6.3%)
Loss of hair	2 (7.7%)	0 (0.0%)
Cardiac symptoms	2 (7.7%)	2 (12.5%)

CTx by FLOT: chemotherapy with 5-fluorouracil, leucovorin, oxaliplatin, and irinotecan * in some patients, more than one symptom occurred.

## Data Availability

The data that support the findings of the study are available upon request from the corresponding author (M.R.).

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
