# Peer review of "Completion of FLOT Therapy, Regardless of Tumor Regression, Significantly Improves Overall Survival in Patients with Esophageal Adenocarcinoma"

_cancers, 2022, doi:10.3390/cancers14041084_

Round 1

Reviewer 1 Report

It is an important topic to study the effects of perioperative FLOT-chemotherapy on esofageal adenocarcinoma in real life material. Could the authors specify:

  • About those 14 patients who did not complete the chemotherapy. How many of them discontinued it in the neoadjuvant and how many in the adjuvant phase? Were those 14 patients older or had they higher ASA or did they differ in some other way compared to those who completed the perioperative chemotherapy?
  • Did the 53 patients who received FLOT complete it also in the adjuvant phase?
  • Did the authors exclude those patients from this study who were started with neoadjuvant FLOT but did not reach surgery?
  • It would be of interest to know 30d and 90d mortality rates (%). Could the authors add those information to table 1?
  • Why have the authors included the patients having imcomplete chemotherapy to minimal response group? Could the results be also presented without having those patients in the minimal response group?
  • Why were the patienst who died during 30d period after the operation excluded from the survival analysis?
  • The authors state that the completion of the systemic therapy regardless of tumor regression grading improved the outcome of patients with esophageal adenocarcinoma. However this might be due to selection bias as the patients in the surgery only group were older and had higher ASA and had postoperative mortality compared to chemotherapy group. The only clear conclusion is that the patients with good response to perioperative chemotherapy benefit from it.

Author Response

Reviewer #1

It is an important topic to study the effects of perioperative FLOT-chemotherapy on esophageal adenocarcinoma in real life material. Could the authors specify:

  1. About those 14 patients who did not complete the chemotherapy. How many of them discontinued it in the neoadjuvant and how many in the adjuvant phase? Were those 14 patients older or had they higher ASA or did they differ in some other way compared to those who completed the perioperative chemotherapy?

We thank reviewer #1 for this good remark. All of the 14 patients received a full cycle of neo-adjuvant chemotherapy and did not finish the adjuvant regime of the therapy. We performed a sub group analysis comparing these 14 patients to the 57 patients, who received complete perioperative chemotherapy. There were no significant differences in regard to age, ASA score, Clavien-Dindo Score and anastomotic leakage rates.

We therefore, added the following sentence to the manuscript at page 3:

“In subgroup analysis, the 14 patients with incomplete adjuvant chemotherapy showed no significant differences in regard to age, ASA score, Clavien-Dindo Score and anastomotic leakage rate (data not shown).”

  1. Did the 53 patients who received FLOT complete it also in the adjuvant phase?

All 57 patients who received complete FLOT also finished the adjuvant phase. To be more specific, we added the following to the method section: “Of the latter group, 57 (80.3%) patients received the full (neoadjuvant and adjuvant) FLOT therapy regimen perioperatively, while 14 (19.7%) patients did not complete the chemotherapy following surgery.

  1. Did the authors exclude those patients from this study who were started with neoadjuvant FLOT but did not reach surgery?

Yes, we excluded all patients that did not reach surgery, since we wanted to investigate the impact of continuation or discontinuation of chemotherapy after esophagectomy. To clarify this issue, we edited the following sentence in the method section: “Only patients with histologically confirmed adenocarcinomas, who subsequently received an oncologic esophagectomy, were selected.”

  1. It would be of interest to know 30d and 90d mortality rates (%). Could the authors add those information to table 1?

Thank you very much for this comment. In the cohort of 124 patients, two patients died during the postoperative course in hospital. These two patients were in the group “surgery only”. One patient died after 14 days postoperatively and the other patient died after 95 days. The data is shown as Clavien-Dindo 5 in table 1.

  1. Why have the authors included the patients having incomplete chemotherapy to minimal response group? Could the results be also presented without having those patients in the minimal response group?

We thank Reviewer #1 for the question since it improves understanding the manuscript. We only included patients in the minimal or no response group after completion of the adjuvant regime. The term incomplete chemotherapy relates to the discontinuation of the chemotherapy after the neoadjuvant therapy and surgery. Hence, only the adjuvant part of FLOT was not administered.  Therefore, the histological response rate can be evaluated properly. We rephrased the Statistical analysis section accordingly: “Surgery alone and patients with surgery and perioperative chemotherapy (FLOT) were compared in terms of overall survival. Different subgroup analyses were performed. The group with perioperative chemotherapy was divided into two groups: completion of neoadjuvant and adjuvant chemotherapy vs. neoadjuvant chemotherapy only“

  1. Why were the patients who died during 30d period after the operation excluded from the survival analysis?

A good remark made by Reviewer #1. This was a mistake in the methods section. All 124 patients were included for survival analyses. We revised the paragraph on page 3 accordingly.

  1. The authors state that the completion of the systemic therapy regardless of tumor regression grading improved the outcome of patients with esophageal adenocarcinoma. However this might be due to selection bias as the patients in the surgery only group were older and had higher ASA and had postoperative mortality compared to chemotherapy group. The only clear conclusion is that the patients with good response to perioperative chemotherapy benefit from it.

We thank reviewer 1 for this significant remark. In our clinical experience, patients tend to be more reluctant to adjuvant chemotherapy after surgery if side effects during the neoadjuvant phase have occurred. Especially, since patients regularly present with a reduced health status after esophagectomies.  Hence by this analysis, we wanted to emphasize the need of completion of chemotherapy in the clinical routine especially, if only a minor or no histological response has occurred.

Reviewer 2 Report

The main problem are the small patient groups, mainly the subgroup of 19 patients. Statistics could be biased by such small numbers. The results should rephrased with some caution. Similarly, this fact should be mentioned in the discussion as shortcoming. 

Perioperative chemotherapy or neoadjuvant radio-chemotherapy is indicated in cT3 and/or cN+ tumors. How could 53 patients treated by surgery alone, if advanced tumor stages were found?

Table 1: If it is correctly depicted that the pT categories are shown, the found difference must be interpreted as down-staging effect of the preoperative chemotherapy.

Table 1: Why are N category is shown in Roman letters? Does the authors meant N1, N2 and N3? Is it pN or cN?

Up to 15% of patients were classified as ASA IV, such patients are normally excluded from major surgery, as e.g. esophagectomy. Please comment.

Patient numbers should be added to the Kaplan-Meier curves.

The method section is incomplete: information on the used complication assessment of surgical and chemotherapy-related complications is lacking.

Among the authors, there is no medical oncologist. This is somewhat surprising, as the aspects regarding the chemotherapy are probably better evaluated.

The authors should better define what it means "completed chemotherapy". Does it means all cycles or complete dosages? Or even all cycles but reduced dosages?

Author Response

Reviewer #2

  1. The main problem are the small patient groups, mainly the subgroup of 19 patients. Statistics could be biased by such small numbers. The results should rephrased with some caution. Similarly, this fact should be mentioned in the discussion as shortcoming.

We agree with Reviewer #2 that a higher number of analyzed patients would be desirable. However, only few patients discontinue adjuvant chemotherapy and hence it is difficult to include morre patients in one esophageal cancer center. 

To put our results in a more cautious context we have added the following to the discussion: “However, our data are of a retrospective nature and the investigated groups especially in patients treated with FLOT and who did not receive adjuvant chemotherapy, are relatively small. Thus, the results need to be interpreted with caution and further prospective multi-center studies are needed to validate our findings.“

  1. Perioperative chemotherapy or neoadjuvant radio-chemotherapy is indicated in cT3 and/or cN+ tumors. How could 53 patients treated by surgery alone, if advanced tumor stages were found?

We thank Reviewer #2 for this remark. Indeed, locally advanced esophageal cancers should be treated with chemotherapy prior to surgery since an improved outcome is expected. However, a high number of patients is not staged correctly. Especially, the local infiltrations and nodal status remain unclear despite a proper staging. In our clinic cT2 cN0 patients do not receive chemotherapy and are treated by surgery only. To address this issue, we added the following passage to the discussion:

“In our cohort, a high number of locally advanced esophageal cancers was found to have received only surgery although a perioperative chemotherapy should have been given. This might be explained by staging errors despite a proper preoperative staging using CT scan, endoscopy and endoscopic ultrasound. Dolan et al. investigated a chemotherapy naïve cohort of cT2cN0 esophageal cancers and identified an understaging in 56% of the patients. Also, a spread of the cancer to the regional lymph nodes was detected in the pathological specimens in 52% underlying the difficulty of correct preoperative staging [1]. In our clinic, cT2 and cN0 staged esophageal cancers are treated by surgery only. However, controversy exists whether cT2 cN0 esophageal cancers should also be treated by chemotherapy since many of the cancers are not staged properly [2,3].“

  1. Table 1: If it is correctly depicted that thepT categories are shown, the found difference must be interpreted as down-staging effect of the preoperative chemotherapy.

Reviewer #2 is correct. We also interpret the difference as a downstaging due to preoperative chemotherapy. Therefore, we have stated the following in the discussion section: “However, we identified a significantly reduced local tumor extension (T category) and lymph node involvement (N category) in the group treated with FLOT. This might be explained by the achieved downstaging of the local tumor burden by the administered neoadjuvant chemotherapy.”

  1. Table 1: Why are N category is shown in Roman letters? Does the authors meant N1, N2 and N3? Is itpN or cN?

We thank Reviewer #2 for this remark and changed the table 1 accordingly. The T- and N- category is now shown as yp/pT and yp/pN and classified as yp/pN 0-3.

  1. Up to 15% of patients were classified as ASA IV, such patients are normally excluded from major surgery, as e.g. esophagectomy. Please comment.

Reviewer #2 is correct about the high number of ASA IV patients receiving surgery. However, as a tertiary center for esophageal cancer surgery we see a relatively high number of patients with significant comorbidities. These patients are interdisciplinary reviewed in-depth and if found to be eligible for surgery also treated with an oncologic esophagectomy. Usually, a neoadjuvant treatment is not given because of the presence of severe comorbidities. This explains the higher number of patients in the surgery only group.

To address this issue, we have added the following to the discussion section: “Of note, 22.6% of the patients presented with an ASA-score of IV. This can be explained by the fact, that as a tertiary center for esophageal cancer surgery a high number of patients with significant comorbidities are presented. These patients are interdisciplinary reviewed in-depth and if found to be eligible for surgery also treated with an oncologic esophagectomy. Usually, a neoadjuvant treatment is not given to these patients because of the presence of severe comorbidities. This explains the higher number of patients in the surgery only group.”

  1. Patient numbers should be added to the Kaplan-Meier curves.

We added the patient numbers to the Kaplan-Meier curves as requested.

  1. The method section is incomplete: information on the used complication assessment of surgical and chemotherapy-related complications is lacking.

Reviewer #2 is correct. We added the appropriate remarks to the method section: “Therefore, the medical record was screened for common side effects of the chemotherapy and complication on an in-patient basis were classified according to Dindo et al. [4].“

  1. Among the authors, there is no medical oncologist. This is somewhat surprising, as the aspects regarding the chemotherapy are probably better evaluated.

Reviewer #2 is right about the missing oncologists. This is an interdisciplinary topic of treatment of esophageal cancer and therefore we invited and included C. Bokemeyer and M. Sinn from the Department of Oncology, University Hospital Hamburg-Eppendorf, Hamburg, Germany to join this study project. Both performed an in-depth review of the results and the manuscript and thereby significantly contributed to the paper.

  1. The authors should better define what it means "completed chemotherapy". Does it means all cycles or complete dosages? Or even all cycles but reduced dosages?

We thank Reviewer #2 for this remark. We indeed looked only at the number of administered cycles and not on the number of reduced dosages, since it would have been not possible to perform a proper statistical analysis.

Hence, we added the following to the method section: “…completion of neoadjuvant and adjuvant chemotherapy (receiving all cycles) vs. neoadjuvant chemotherapy only…“ and the following to the discussion section: “Also, there might be a heterogeneity in patients treated with FLOT therapy, since we only investigated the number of administered cycles of chemotherapy and did not analyze the effect of a dosage reduction. Hence, the results need to be interpreted with caution and further prospective multi-center studies are needed to validate our findings.”

Round 2

Reviewer 2 Report

Regarding the patients that were classified ASA IV: By only operating such patients, you introduce a selection bias. As they were excluded from preoperative chemotherapy due to their co-morbidities, they were automatically also not eligible for postoperative chemotherapy. 

By the way, is it really true that the operative risk for ASA IV patients is less than the risk for adverse effects of preoperative chemotherapy? You must show this evidence.

Exclusion of these patients should considered. 

Author Response

We would like to thank you for the positive review feedback. In the following, you will find point-by-point responses with regard to the reviewer comments.

Reviewer 2:

Regarding the patients that were classified ASA IV: By only operating such patients, you introduce a selection bias. As they were excluded from preoperative chemotherapy due to their co-morbidities, they were automatically also not eligible for postoperative chemotherapy. By the way, is it really true that the operative risk for ASA IV patients is less than the risk for adverse effects of preoperative chemotherapy? You must show this evidence. Exclusion of these patients should considered.

We thank reviewer 2 for this good remark. Patients classified as ASA IV have severe comorbidities and tend to have poor oncological outcome in the literature. However, these patients belong to daily clinical routine in a university hospital and we wanted show “real life” data of our patients’ cohort. This study included 18 patients (14.5%) classified as ASA IV. Six patients of these underwent perioperative chemotherapy and 12 patients underwent only surgery due to staging inaccuracy or comorbidities, which made chemotherapy impossible (see table 1 in the manuscript). There were no patients, who were excluded automatically from chemotherapy due to ASA IV. (see discussion part, page 11). Thus, there is no selection bias in this cohort. However, for further evidence, we performed subgroup analyses of the ASA IV patients. Interestingly, there was no significant difference in completion of perioperative chemotherapy in these patients (p=0.299) and there were no significant differences in complications due to chemotherapy (p=0.235) in comparison to patients with ASA I-III.

Further, the aim of this study was to evaluate the impact of complete neoadjuvant and adjuvant chemotherapy in patients with adenocarcinoma of the esophagus in comparison to incomplete perioperative chemotherapy. The “surgery only” group is only a historical baseline cohort to show the benefit of perioperative chemotherapy.  

We totally agree to reviewer 2 that the operative risk for ASA IV patients is not less than the risk for adverse effects of preoperative chemotherapy. Thus, we try to offer every a patient a preoperative chemotherapy according to the pre-treatment staging.  In conclusion, we think that it is useful to include the patients classified as ASA IV in this study because they represent “real life” data.

Sincerely,

Matthias Reeh

Round 3

Reviewer 2 Report

no further comments